# Two-Dimensional Transmission of Four-Dimensional LDPC-Coded Modulation with Slepian Sequences for DSP-Free 40 km Metro Network Applications

**DOI:** 10.3390/s22051815

**Published:** 2022-02-25

**Authors:** Xiao Han, Ivan B. Djordjevic

**Affiliations:** Department of Electrical and Computer Engineering, University of Arizona, 1230 E Speedway Blvd, Tucson, AZ 85721, USA; ivan@arizona.edu

**Keywords:** 4D modulation, Slepian sequence, DSP-free, 16QAM, metro network, 2D transmission

## Abstract

The growing data demands are pushing researchers to pay more attention to spectrally efficient modulation formats. The four-dimensional (4D) signal constellation modulation format has been investigated for metro networks’ applications to achieve better power efficiency. To cope with such modulation formats, the requirement of better digital signal processing (DSP) is also increasing rapidly. More complicated DSPs bring us extra costs; thus, the DSP-free coherent receivers are also investigated because of the high-power consumption of conventional DSP-based receivers, but the transceivers upgrading also results in extra costs. In this invited paper we implement a 4-dimentional modulation format based on Slepian sequences. We applied LDPC coding and experimentally investigated the BER performance in a two-dimensional (2D) 40 km fiber link transmission and demonstrate that being error free is possible without employing the complicated DSP. We compared our proposed modulation scheme with regular 16QAM and found it outperforms 16QAM with DSP over back-to-back transmission by 3.8 dB improvement in OSNR when BER = 10^−5^, while over 40 km metro network communication link our proposed 4D modulation signals are still successfully transmitted, and the LDPC-coding still works properly with such a new transmission strategy. On the other hand, DSP-free transmission of LDPC-coded 16-QAM exhibits an early error floor phenomenon.

## 1. Introduction

The never-ending demands for higher data rates always exist in a variety of industry fields, which require significant efforts to satisfy the growing data demands and to deal with the incoming bandwidth capacity crunch [1,2,3], pushing researchers to pay more attention to spectrally efficient modulation formats [4]. For high data-rate transmission, the coherent communication has already become the standard communication technique in metro and long-haul networks and is becoming more and more widely applied into the datacenter space. The coherent optics has been used as a key technology for 400 G and higher data rate transmission [5]. Quadrature amplitude modulation (QAM) formats have been investigated in [6,7] to have an excellent performance in achieving higher spectral efficiency with the same forward error correction (FEC) overhead [8]. Compared with the widely used QAM modulation formats in coherent optical communications, the more complex modulation formats, such as 4-dimensional signaling [9,10,11,12], can offer us better power efficiency, by increasing the Euclidean distance between constellation points for a given signal power when compared to the 2D signal space. The 4D modulation formats applications are investigated mainly for long-haul transmission [13], but have also been demonstrated to have advantages for metro networks applications in our previous work [10]. DWDM is an efficient way to increase overall capacity by allocating a huge amount of data to different wavelength channels and transmit them simultaneously [14,15,16,17,18,19]. The application of pulse shaping scheme, together with the DWDM system, can offer us better spectral efficiency [19]. FEC coding is an efficient method to increase the channel performance. Researchers have investigated the bit error rate (BER) performance with different codes [20,21] and found the low-density parity-check (LDPC) coding outperforms other coding schemes in the high code rate region.

The capacity of DWDM optical transmission system can be increased significantly with the application of digital signal processing (DSP). To enable advanced modulation formats, the requirement of better DSP is also increasing rapidly [22]. The phase recovery algorithms for the complicated modulation formats are already investigated in [23,24], but more complicated DSPs bring us extra costs, and the conventional DSP-based receivers require high-power consumption. While the demands of different modulation formats applications are increasing, a DSP application-specific integrated circuit (ASIC) is also required to support multi-modulation and multi-rate formats [25]. Researchers have already investigated DSP-ASIC to support QAM modulation formats with different constellation points for different data rate transmissions [25,26,27]. The constellation shaping scheme, both probabilistic shaping (PS) and geometric shaping (GS), are broadly applied to reduce the gap of channel capacity to Shannon limit, which also require specified DSP scheme to optimize the performance [28,29]. These increasing demands of DSP costs have led researchers to pay more attention to the DSP-free coherent receivers [30,31,32,33]. On the other hand, the terabit optical Ethernet technologies will be affected by security issues, and to solve such problem, the properly designed fiber Bragg gratings (FBGs) application in impulse responses derived from mutually orthogonal Slepian sequences has been advocated in [1] to enable positive rate to convert optical communications [2,3].

In this invited paper, we apply the Slepian sequences-based basis functions to a 16-points 4D modulation format, combined with the LDPC coding for 40 km metro network transmission. We find that with the application of Slepian sequences, the 4D modulation format can be implemented using an I/Q modulator and recovered on receiver side without any high power demanded DSP. The LDPC coding works properly with such a new modulation scheme. We experimentally demonstrate that DSP-free LDPC-coded SS-based 4D modulation scheme significantly outperforms the corresponding LDCP-coded 16-QAM counterpart in both back-to-back configuration and after 40 km transmission.

This invited paper is organized as follows. In Section 2 we describe the proposed SS-based LDPC-coded 4D modulation scheme. Experimental setup and experimental results are provided in Section 3. The last section summarizes the paper.

## 2. Proposed Slepian Sequences-Based 16-4D LDPC-Coded Modulation Scheme

The demands for using higher order and more complex modulation formats are increasing rapidly to satisfy never ending data requirements. To meet such transmission formats needs, more advanced DSP upgrading is required, which will bring us a large amount of implement costs. The Slepian sequence (SS), which is a discrete prolate spheroidal sequence, can offer us some new approaches to reduce DSP complexity and improve both BER performance and spectral efficiency without extra expensive hardware costs. The Slepian sequences $\left\{s_{n}^{\left(j\right)}\left(N,W\right)\right\}$ of the *j*-th order are defined as a real-valued solution to the following system of discrete Equation (1):(1)$$\overset{N-1}{\underset{i=0}{\sum }}\frac{\sin2\pi W\left(n-i\right)}{\pi \left(n-i\right)}s_{n}^{\left(j\right)}\left(N,W\right)=\mu_{j}\left(N,W\right)s_{n}^{\left(j\right)}\left(N,W\right);\;n\in N$$
where *i* and *n* denote the particular sample in each Slepian sequence, *N* is the sequence length, and *j* denotes the particular sequence order out of the set of sequences. The shaping factors $\mu_{j}\left(N,W\right)$ are ordered eigenvalues of the systems of Equation (1) corresponding to the concentration of each SS within the desired time interval of length *N*; thus $0<\mu_{j}\leq 1$, with 1 occurring when the sequence energy is entirely included in the desired time interval. *W* is a discrete bandwidth [1,3].

We can easily generate the Slepian sequences of different orders, which are orthogonal to each other and as such they can be used as a new degree-of-freedom. Figure 1 shows the first 4 orders of the Slepian sequences. The Slepian Sequence we used is aimed to recover the transmitted sequence only with very simple DSP, to efficiently reduce the cost and complexity of the system.

Our proposed Slepian sequence-based 4D LDPC-coded transmitter is shown in Figure 2. A pseudorandom binary sequence (PRBS) is generated and split to 4 parallel sequences as our 4D signal. After the LDPC encoder, the encoded 4 parallel sequences are mapped to our 4D constellation points. The 4D constellation points take values of ±1 for every dimension. The corresponding output is then sent to the Slepian sequence mapping module, and finally we impose the 4D modulated signal on 2D I/Q modulator, by mapping the first two coordinates to the in-phase (I) channel and the last two coordinates to the quadrature (Q) channel of an electro-optical I/Q modulator.

As an example, to transmit a constellation point of (+1, +1, +1, −1), we first map the symbol to a group of Slepian sequences:{+(1st order), +(2nd order), +(3rd order), −(4th order)},
then we have the first and second order Slepian sequences mapped to the I-channel, while the third and fourth order Slepian sequences mapped to Q-channel as follows:$$\{\begin{array}{c} I=\left(1\mathrm{st}\;\mathrm{order}\right)\;\mathrm{and}\;\left(2\mathrm{nd}\;\mathrm{order}\right)\;\\ Q=\left(3\mathrm{rd}\;\mathrm{order}\right)\;\mathrm{and}\;\left(4\mathrm{th}\;\mathrm{order}\right) \end{array}$$

## 3. Experimental Setup and Experimental Results

The experimental setup is shown in Figure 3. The SS 16-4D LDPC-coded generator is introduced before in Figure 2, the output sequences are sent to a Keysight arbitrary waveform generator (AWGen) with sampling rate of 120 GSa/s. Then, the analog signals are sent to an I/Q modulator. The laser source we used is a 10 kHz-linewidth, continuous-wave, tunable source with center frequency of 193.4 THz. The optical signal after the I/Q modulator is amplified by an erbium-doped fiber amplifier (EDFA) of 6 dB noise figure. The boosted signals are coupled with the amplified spontaneous emission (ASE) noise by a 2 × 2 3 dB coupler. We employed a variable optical attenuator (VOA) after the ASE noise source to emulate different optical SNR (OSNR) and then send the signal to the 40 km fiber link for the metro network communication. At the receiver side, we first applied a tunable filter (TF) to obtain the signals with our target frequency, and after amplifying the signal with EDFA the signals are sent to the coherent receiver, with the integrated coherent receiver (ICR) board and a 100 GS/s sampling rate oscilloscope from Tektronix, followed by the correlation/matched filter decision circuitry. After the correlation detector we also use the correlation approach to figure out the beginning of the LDPC-coded block (FEC frame). Then, we sent the resulting projections along the SS basis functions to the LDPC decoder to analyze the post-BER performance.

The structure of our correlation detector is shown in Figure 4. Both real and imaginary parts of the received signals are split and each branch is multiplied by each order SS to calculate the correlation, followed by summation operation to implement the integration operation. We also figured out the beginning of the LDPC-coded block before the splitter. Then, we sent the resulting projections along the SS basis functions to the LDPC decoder to analyze the post-FEC BER performance. We can see the only DSP we implemented is just the correlation calculation, which strictly speaking does not belong to the DSP, and we use only 3–5 pilot symbols to perform the group sign recovery.

The structure of our DSP progress is shown in Figure 5. The received sequences are collected from the real-time oscilloscope (Tektronics with sampling rate of 100 GSa/s). Given that the sampling rates of AWGen and oscilloscope are different, we need to do the resampling first. After that we use the correlation calculation to do the synchronization, to find out the starting point of the codeword. The algorithm we used for the equalization is the least mean squares (LMS) algorithm. Then, we do the phase recovery, using the 4th power algorithm. After this DSP progress, we normalized the signal and then calculate the log-likelihood ratios (LLRs) for LDPC decoding.

The transmitted and received waveforms in a back-to-back (b2b) configuration are shown in Figure 6 with 250 samples. The red curve represents our generated sequence of I channel, while the blue curve is our received sequence without any noise loaded for I channel. We can see our sequence is recovered efficiently.

The pre-FEC and post-FEC BER performances of SS-4D modulation and the traditional 16QAM are summarized in Figure 7a,b. We used both 120 GSa/s and 92 GSa/s Keysight AWGens to implement the 32 samples and 24 samples per Slepian sequence, respectively. The baud rate is 120 G/32 = 3.75 GBaud for 32 samples and 92 G/24 = 3.833 GBaud for 24 samples, respectively; while the baud rate of 16QAM is also 3.75 GBaud. The length of the codeword we used for both SS and 16QAM is 5648 bits, with 4239 information bits and 1409 parity check bits, thus, the code rate is 0.75. The red curves represent the BER performance of the 4D modulated Slepian sequences with 32 samples; the blue curves are the BER performances of 4D modulated Slepian sequence with 24 samples; the cyan plots show the BER performance of regular 16QAM modulation after DSP; and non-DSP 16QAM BER performance is illustrated in the green plot. The solid plots represent the post-LDPC BER performance, while dashed curves are for pre-LDPC BER performance.

The BER performance corresponding to the back-to-back configuration is shown in Figure 7a, we can see the proposed 4D modulation signals are successfully transmitted using a 4D modulator with the SS-based basis functions and the proposed LDPC-coded SS-based 4D modulation scheme significantly outperforms the regular LDPC-coded 16QAM modulation format with DSP, achieving 3.8 dB and 3.4 dB gain in OSNR when BER = 10^−5^ for 32 samples and 24 samples, respectively.

Figure 7b shows the BER performance of different modulation formats after 40 km transmission. Obviously, without DSP the LDPC-coded 16-QAM exhibits early error floor and any reliable transmission is not possible. The DSP-free LDPC-code SS-based 4D modulation scheme is able to achieve error-free transmission. On the other hand, we can clearly see that the regular LDPC-coded 16QAM with DSP now outperforms the LDPC-coded DSP-free 4D SS signals for about 2.2 dB in OSNR at BER = 10^−5^, but this comparison is not fair. If we compare our results of back-to-back performance with those of 40 km link transmission, we can find that with the DSP, regular 16QAM does not have too much performance loss after 40 km transmission, but the performance of our proposed DSP-free 4D SS signal deviated a lot after 40 km transmission. This is because we are using the phase to carry information, but we do not apply any phase recovery. Researchers in [34,35,36] have already put efforts to the DSP-free transmission for simple modulation formats, such as PAM4, which is non-phase-sensitive, along with a very short transmission distance. If we use different orders of waveforms to transmit all information, the performance could be improved. In [36], the researchers implemented a DSP-free transmission without equalization. If we can try to apply some simple phase recovery, the performance will be promisingly improved.

## 4. Concluding Remarks

In this invited paper, we have applied the Slepian sequence in DSP-free 4D signaling in back-to-back configuration and in 40 km link transmission. We have investigated the effect of varying the number of sample points per waveform. We have compared the BER performance of the proposed SS-based LDPC-coded 4D scheme with regular LDPC-coded 16QAM modulation formats and find that the proposed DSP-free 4D SS signaling significantly outperforms DSP-free 16QAM modulation format in both b2b configuration and after 40 km transmission.

The proposed Slepian sequences-based 4D LDPC-coded modulation scheme is suitable for metro applications without any DSP. The proposed LDPC-coded SS-based 4D modulations scheme can also be used in data center applications, with results similar to Figure 7a.

To improve the baud rates instead of using AWGens to generate Slepian sequences we should fabricate the corresponding waveguide Bragg grating devices in a similar fashion as described in ref. [37].

## Figures and Tables

**Figure 1 sensors-22-01815-f001:**
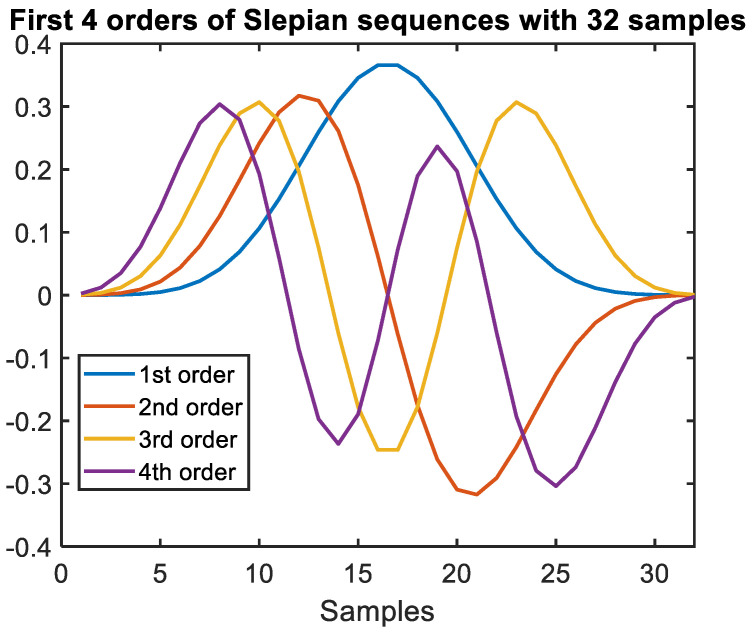
Different orders of Slepian sequences with 32 samples/waveform used in experimental verification.

**Figure 2 sensors-22-01815-f002:**
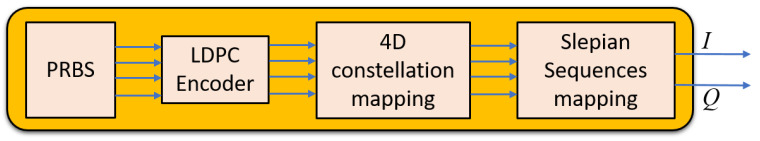
Slepian sequences-based 4D LDPC-coded transmitter.

**Figure 3 sensors-22-01815-f003:**
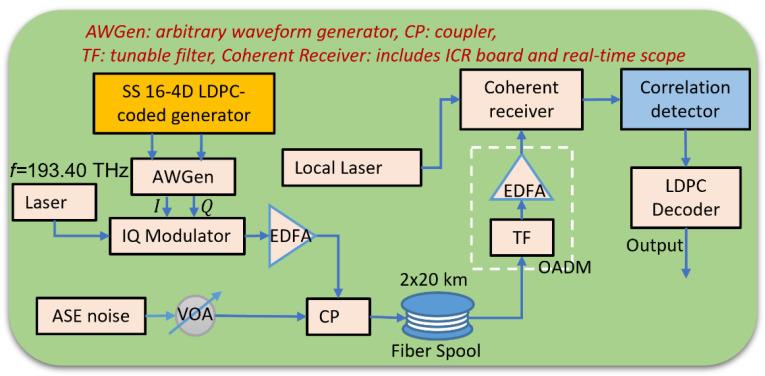
Experimental setup for SS-based 16-4D LDPC coded modulation scheme. OADM: optical add-drop multiplexer emulator.

**Figure 4 sensors-22-01815-f004:**
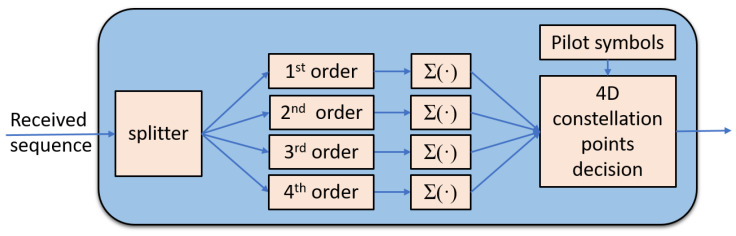
The configuration of the correlation detector.

**Figure 5 sensors-22-01815-f005:**
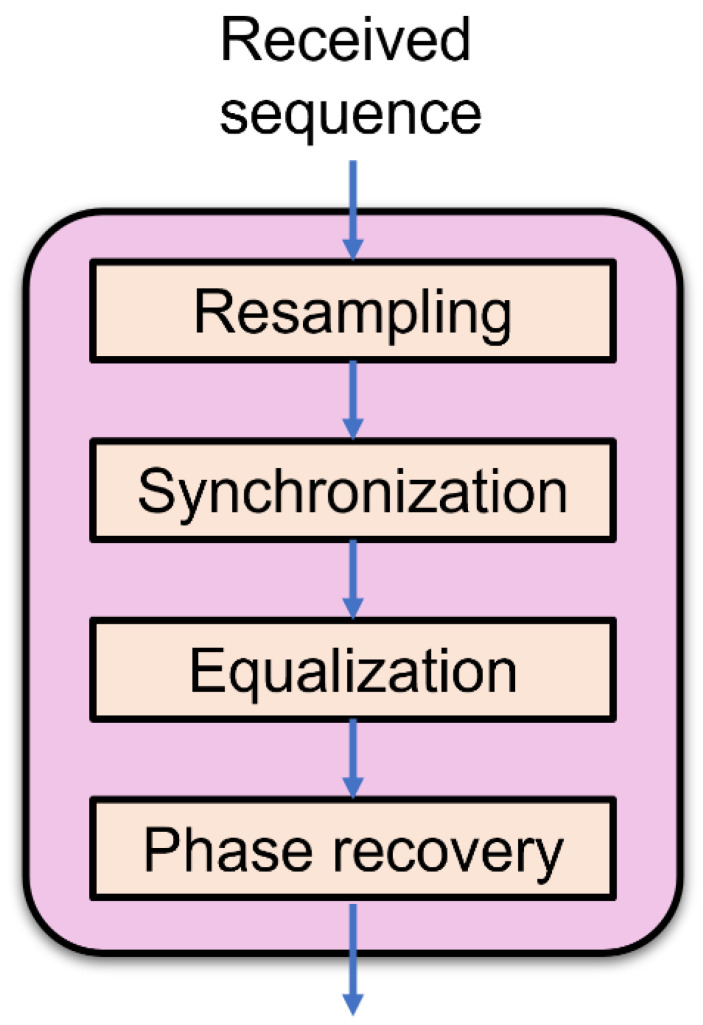
The configuration of the DSP progress for 16QAM.

**Figure 6 sensors-22-01815-f006:**
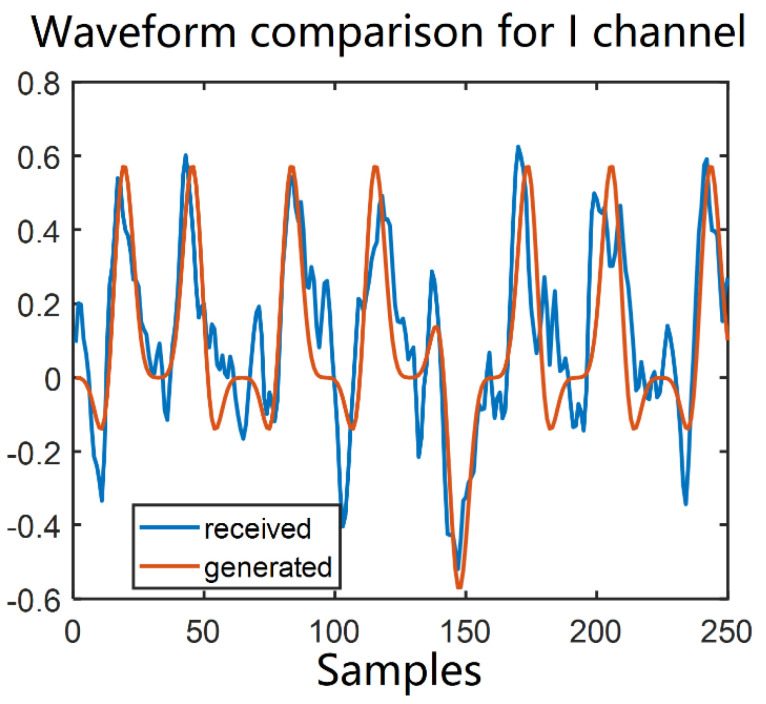
Comparison of received and transmitted waveforms.

**Figure 7 sensors-22-01815-f007:**
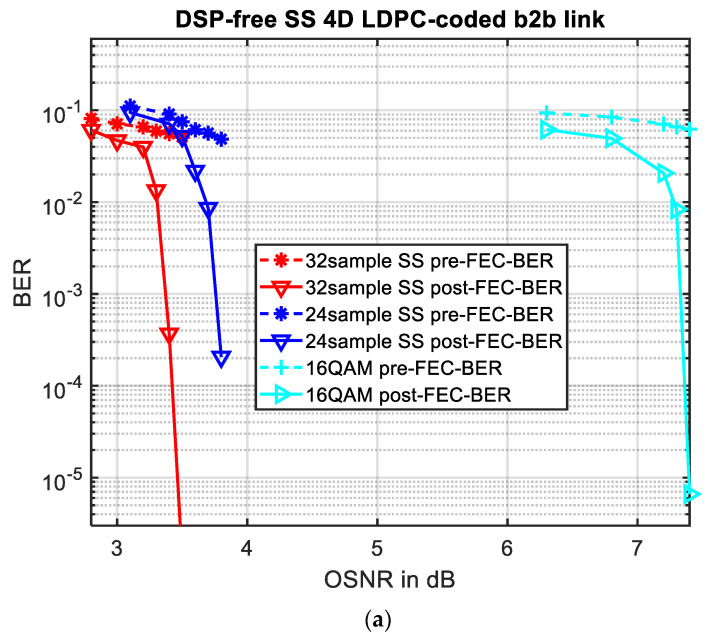
BER performance over OSNR for 32 and 24 samples SS 4D modulation and regular 16QAM with and without DSP: (**a**) in the back-to-back configuration and (**b**) after 40 km SMF link transmission.

## Data Availability

Not applicable.

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
