# Peer review of "Two-Dimensional Transmission of Four-Dimensional LDPC-Coded Modulation with Slepian Sequences for DSP-Free 40 km Metro Network Applications"

_sensors, 2022, doi:10.3390/s22051815_

Round 1
Reviewer 1 Report
This manuscript proposed a two-dimensional (2D) transmission of four-dimensional (4D) LDPC-coded modulation with Slepian sequences for an optical communication systems that can be used for metro networks with the length of 40 km. The proposed method can help reduce the dependence on DSP hardware. The proposed method is tested by laboratory experiments
Overall, the manuscript is well-written and well-organized and the proposed method is novel and good technically. The reviewer has some minor comments to improve the quality of the manuscript:
- The authors should provide more references to survey on the alternative methods to clarify the benefits of the proposed one. In particular, why are Slepian sequences selected for the proposed method? What are the alternative methods rather than Slepian sequences and what are their disadvantages?
- The parameters of LDPC codes used in this manuscript should be described clearly (codeword length, generating matrix, code rate, etc.) to evaluate the complexity of the transceiver processing.
- The authors claim that the proposed method is DSP-free, but I think the LDPC encoder and decoder may require a significant processing load. Please justify this issue.
- The manuscript is well-written generally, but there is still some room for improvement. In particular, there are several duplications in the manuscript:
+ "pushing researchers to ..." --> repeated at least 3 times
+ "on the other hand, ..." --> repeated a lot of times, even in 2 consecutive sentences.
- At line 133 (page 4), there is a mention of "Figure 5(a)", but there is only Figure 5 in the manuscript (no (a), (b) ,...).
Author Response
- The authors should provide more references to survey on the alternative methods to clarify the benefits of the proposed one. In particular, why are Slepian sequences selected for the proposed method? What are the alternative methods rather than Slepian sequences and what are their disadvantages?
A: This is the first time to apply Slepian sequence (SS) into optical communication, so there is not too much reference for the SS, except for those already cited. In this paper we study the feasibility of SS application as a new degree-of-freedom. The disadvantage is for our current experiment we can’t apply a high baud rate, and this is also discussed in the concluding remarks section. An alternative would be employing the spatial modes, but this is optical domain.
- The parameters of LDPC codes used in this manuscript should be described clearly (codeword length, generating matrix, code rate, etc.) to evaluate the complexity of the transceiver processing.
A: The details related to the codeword length and code rate are added.
- The authors claim that the proposed method is DSP-free, but I think the LDPC encoder and decoder may require a significant processing load. Please justify this issue.
A: The DSP is applied to get the pre-LDPC BER, the LDPC encoder and decoder are not regarded as a part of the DSP progress.
- The manuscript is well-written generally, but there is still some room for improvement. In particular, there are several duplications in the manuscript:
+ "pushing researchers to ..." --> repeated at least 3 times
A: Modified, thank you for pointing out.
+ "on the other hand, ..." --> repeated a lot of times, even in 2 consecutive sentences.
A: Modified, thank you for pointing out.
- At line 133 (page 4), there is a mention of "Figure 5(a)", but there is only Figure 5 in the manuscript (no (a), (b) ,...).
A: Modified, thank you for pointing out.
Reviewer 2 Report
The authors consider the problem of DSP-free optical communication that makes use of a 4-dimensional modulation format based on Slepian sequences. The authors demonstrate that is possible to transmit error-free without requiring complicated DSP and also avoiding the high power consumption of the conventional DSP. The topic is current while the novelty of the proposed method is limited.
Section 2 can be introduced including an introduction and not beginning immediately with Slepian sequences. Also, axes' names and titles need to be included in figure 1 and also in figure 5.
Finally, in the introduction section, more references can be used as 30% of the included references are based on the authors' previous work.
Author Response
A: Contents in section 1 and 2 is expanded, Figures 1 and 5 are updated, additional references are added.